# Membership Inference Attacks on Lottery Ticket Networks

**Aadesh Bagmar** [* 1]   **Shishira Maiya** [* 1]   **Shruti Bidwalkar** [* 1]   **Amol Deshpande** [1]

## Abstract

The vulnerability of the Lottery Ticket Hypothesis has not been studied from the purview of Membership Inference Attacks. Through this work, we are the first to empirically show that the lottery ticket networks are equally vulnerable to membership inference attacks. A Membership Inference Attack (MIA) is the process of determining whether a data sample belongs to a training set of a trained model or not. Membership Inference Attacks could leak critical information about the training data that can be used for targeted attacks. Recent deep learning models often have very large memory footprints and a high computational cost associated with training and drawing inferences. Lottery Ticket Hypothesis is used to prune the networks to find smaller sub-networks that at least match the performance of the original model in terms of test accuracy in a similar number of iterations. We used CIFAR-10, CIFAR-100, and ImageNet datasets to perform image classification tasks and observe that the attack accuracies are similar. We also see that the attack accuracy varies directly according to the number of classes in the dataset and the sparsity of the network. We demonstrate that these attacks are transferable across models with high accuracy.

## 1. Introduction

Machine learning models are integrated into a wide range of applications today, including ones involving critical, sensitive, or confidential data. Despite the tremendous advantages of these models, they are vulnerable to various kinds of attacks that compromise the efficacy [cite adv] and the privacy (Shokri et al., 2017) of the system and the underlying data. In Shokri et al. (2017), the authors designed a membership inference attack (MIA) which made it possible

to recover the individual records from training data, raising major privacy concerns. They exploit the fact that most models tend to overfit and have major differences in confidence values of samples belonging to training and test splits to create a "shadow model" that distinguishes between them. Lottery Ticket Hypothesis (Frankle & Carbin, 2018) is a breakthrough in the area of neural network pruning. It shows the existence of sparse sub-networks that at least match the test accuracy of the original dense network and train in at most similar number of iterations. These sub-networks that are composed of the "winning lottery ticket" have advantages of a smaller memory footprint and lesser compute resources needed for inference while still maintaining the test accuracy.

A natural question then is, are such "lottery ticket networks", because of their ability to generalize and not overfit to the same extent, less susceptible to membership inference attacks? In this paper, we attempt to answer that question empirically.

Our main contributions include performing experiments to evaluate the impact of membership inference attacks on lottery ticket networks. We analyse the performance of ResNet18 and ResNet50 classifiers trained on various datasets with and without lottery tickets. We use a single shadow attack model approach and employ a simple classifier (Multi-Layer Perceptron). We observe that the accuracy and precision of our attacks increases with increase in the number of classes and more importantly, *we show that the lottery ticket networks are equally vulnerable to membership inference attacks as the original dense networks.*

Rest of the paper is organized as follows. We first briefly describe the key concepts of membership inference attacks and lottery tickets. We then describe our experimental setup and report the results. Finally, we discuss the next steps and provide some future directions for our work.

## 2. Background

In this section, we briefly describe the background required to understand our work.

---

[*]Equal contribution  [1]Department of Computer Science, University of Maryland, College Park, Maryland, USA. Correspondence to: Aadesh Bagmar <aadesh@umd.edu>.

*Accepted by the ICML 2021 workshop on A Blessing in Disguise: The Prospects and Perils of Adversarial Machine Learning.* Copyright 2021 by the author(s).

## 2.1. Membership Inference Attacks

Membership inference attacks (MIA) aim to identify whether a data sample was used to train a machine learning model or not. These attacks have been successfully carried out on centralized supervised learning and unsupervised learning models and also distributed learning based Federated Learning models (Hu et al., 2021).

These attacks work even if the attacker does not have access to the original training data that was used to train the target model. Shokri et al. (2017) describe a method wherein they train multiple "shadow models" that mimic the behaviour of the target model. This is a type of a white-box attack where the architecture of the targeted model and the training dataset membership of this shadow model is known. Salem et al. (2018) showed that a single shadow network is sufficient too.

Membership inference attacks have been studied extensively (Shokri et al., 2017; Nasr et al., 2018; Li & Zhang, 2020) and across different domains (Danhier et al., 2020; Salem et al., 2018; Liu et al., 2019; He et al., 2020). Different types of attacks including neural network based and metric based have been proposed and researchers have shown successful black box and white box approaches. Defenses against such attacks have been studied as well and mostly focus around reducing overfitting and reducing the influence of certain data points. Nasr et al. (2018) suggest using adversarial regularization training to defend against this. Shokri et al. (2017) suggested defense techniques like restricting the prediction vector to top $k$ classes; however, highly accurate attacks are still possible even when the model reveals minimal information (Li & Zhang, 2020).

## 2.2. Lottery Ticket Hypothesis

Lottery Ticket Hypothesis shows that it is possible to reduce the parameter counts of trained neural networks by over 90% while still maintaining the accuracy by using efficient pruning techniques. This theory states that in dense, randomly-initialized feed-forward networks there exist such sub-networks which when trained in isolation reach test accuracy comparable to the original network in a similar number of iterations (Frankle & Carbin, 2018). To identify these sparse sub-networks, the authors aim to prune the weights with the smallest-magnitude using Iterative Magnitude Pruning.

Since then there has been follow up work to further analyse the hypothesis (Frankle et al., 2019; Malach et al., 2020; Frankle et al., 2020a; Morcos et al., 2019) and apply this hypothesis to various domains such as object detection (Girish et al., 2020), for graph neural networks (Chen et al., 2020), for reinforcement learning and NLP (Yu et al., 2019), for pre-trained BERT model (Chen et al., 2021), and so on.

## 3. Experimental Setup

Our goal is to understand if using lottery ticket networks has any impact on privacy of the data used for training the model, given that lottery ticket networks are smaller and more generalizable. We do this by empirically comparing privacy leakages in regular neural networks against privacy leakages in lottery ticket networks.

---

**Algorithm 1** One Shot Pruning

---

1: Randomly initialize network $f$ with initial weights $w_0$, mask $m_0 = 1$, prune target percentage $p$
2: Train network for N iterations $f(x; w_0) \rightarrow f(x; w_n)$
3: Prune bottom $p\%$ of $w_n$ by magnitude to obtain $m_n$
4: Reset to initial weights $w_0$
5: Retrain pruned network for N iterations $f(x; m_n \odot w_0) \rightarrow f(x; m_n \odot w_p)$

---

### 3.1. Datasets

We conducted our experiments on CIFAR-10, CIFAR-100 (CIF) and ImageNet datasets. We use models trained on the ResNet18 architecture unless mentioned otherwise.

### 3.2. System Setup

#### 3.2.1. OBTAINING LOTTERY TICKETS

The first requirement of our experiments is lottery tickets at varying levels of sparsity across different datasets. For this, we utilize a variant of Iterative Magnitude Pruning (IMP) algorithm used in (Frankle & Carbin, 2018) called "One shot pruning" where the number of pruning rounds to obtain desired sparsity is reduced to one. The detailed algorithm is presented in 1.

#### 3.2.2. TRAINING SHADOW MODELS

We generated a dataset to train an attack model using the following steps. We follow these steps for evaluating both, a normal Neural Network and a Lottery Ticket Network created in a similar way as discussed in Frankle et al. (2020b).

- We pass a subset of images which have been used to train our initial network and generate a set of confidence vectors $P(y|x)$. We choose equal number of samples which the model has seen (coming from the original model's training set) and those that the model hasn't seen (from original model's testing set). This is the dataset we use to train our attack model.

- Once we have the confidence vectors, we train a classifier to identify whether a sample belongs to the train set or the test set. This is our shadow model for membership inference.

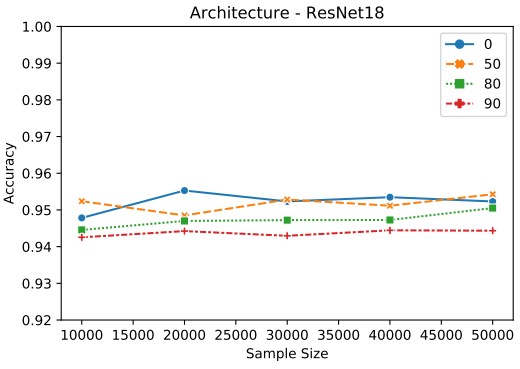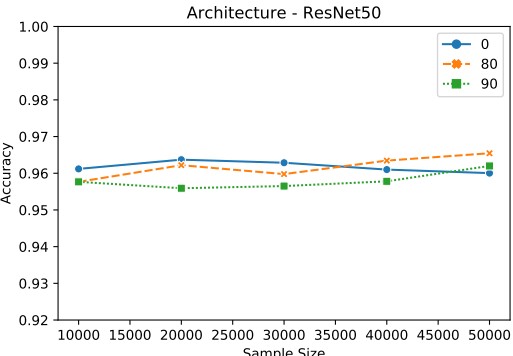

*Figure 1.* Performance of lottery tickets by varying sparsities for Resnet-18 and ResNet-50. Sparsity 0 indicates the original network.

We chose the MLP (Multi-layer Perceptron) classifier because it gave the highest accuracy (0.94) among all the classifiers we tried [SVM(0.89), Random Forests (0.88)]. The MLP classifier optimizes the log-loss function using the stochastic gradient descent. We use the standard implementation from scikit-learn.

## 4. Results and Discussions

In this section, we discuss the results of our experiments.

### 4.1. Primary Analysis

We observe that the accuracy and precision for membership inference attacks on both our baseline model and its corresponding lottery ticket network are almost similar. We show our results across various datasets and models in Table 1. We observe minute differences after 20k samples in the accuracy of our attack model. The attack model trained on Image Net showed high recall as well.

#### 4.1.1. ATTACK EFFICIENCY AND NUMBER OF CLASSES

Our results are similar to those reported by Shokri et al. (2017) where they observe an increase in accuracy with an increase in the number of classes. This is because the attack model receives sparser data as the number of classes increase. We observe that the trend translates to lottery ticket networks as well where we observe an accuracy of 0.5, 0.74 and 0.94 for datasets having 10, 100 and 1000 classes.

#### 4.1.2. ATTACK ACCURACY AND MODEL SPARSITY

We evaluate how attack accuracy varies based on network sparsity (Frankle & Carbin, 2018). We observe similar levels of accuracy for different sparsities when trained on ImageNet ranging between 0.94 and 0.96 for ResNet18 and between 0.95 and 0.97 for ResNet50. We show our results in

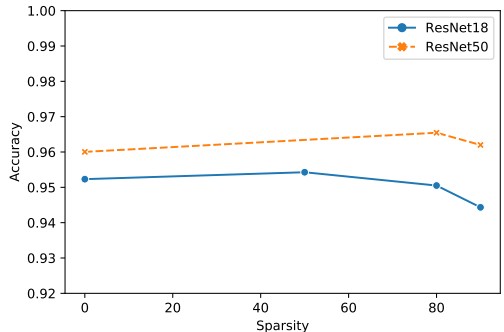

*Figure 2.* The above figure shows the accuracy of our membership inference attack and how it varies if we change the lottery ticket sparsity. We fix the number of training samples to 50,000. We compare results across ResNet18 and ResNet50

Figure 1. We observe a trend of decrease in attack accuracy at very high levels of sparsity.

### 4.2. Comparing different architectures

We wanted to understand if changing the architecture leads to a change in accuracy. We observe a higher accuracy in our attacks in ResNet50 compared to ResNet18 implying that ResNet50 may be more vulnerable. We also observe that the attack accuracy first increases with increase in sparsity and then drops. This is coherent with the results observed by Girish et al. (2020) where they show an increase in accuracy with increase in sparsity followed by a steep drop at higher sparsity levels. We show our results in Figures 2 and 3.

### 4.3. Do membership attacks transfer?

In this section evaluate the efficacy of membership inference attack when the shadow models are trained using the outputs of one model and then used to attack an unknown model. We consider the case where shadow models are obtained by

| | Accuracy Score of MIA (Using MLP) | | Precision Score | |
|---|---|---|---|---|
| Dataset | Baseline Model | Lottery Ticket Network | Baseline Model | Lottery ticket Network |
| CIFAR-10 | 0.503 | 0.504 | 0.57 | 0.59 |
| CIFAR-100 | 0.744 | 0.744 | 0.97 | 0.94 |
| ImageNet 20k samples | 0.944 | 0.940 | 0.945 | 0.931 |
| ImageNet 60k sample | 0.9499 | 0.942 | 0.96 | 0.956 |
| ImageNet 100k samples | 0.952 | 0.944 | 0.96 | 0.957 |

*Table 1.* **Accuracy and Precision of MIA for different datasets**

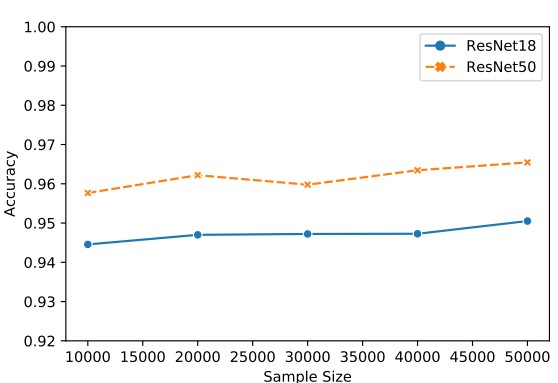

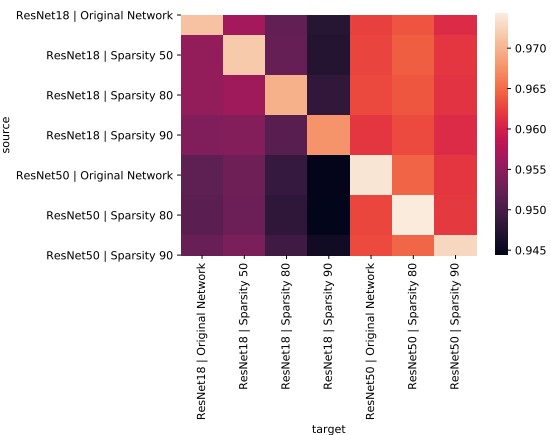

*Figure 3.* The above figure shows performance of our membership inference attack and how it varies if we change the sample size used for training. We fix the network sparsity to 80%. We compare results across ResNet18 and ResNet50

*Figure 4.* The above heat-map shows the accuracy with which attacks are transferable to other models. Source denotes the model whose outputs were used to train an attack model and the target is the model which is attacked. Thus, the diagonal shows our base case where the source and the target are the same.

training on confidence outputs from Resnet-18 and Resnet-50 architectures on Imagenet. These are the observations from these set of experiments:

- We can clearly see that the highest attack success are when we attack the same model, as shown in the diagonal of 4.

- By observing the columns of the heatmap across the two architectures, there appears to be slight *decrease* in the efficacy of shadow model as we *increase* sparsity

- We can also see that the attack is more successful on Resnet-50 models compared to Resnet-18.

## 5. Conclusion and Future Work

Our results seem promising and open future research directions. We observed that the lottery ticket models are as vulnerable as their original counterparts and we observe minute differences in accuracies. We also see that the attacks

vary based on number of classes in the dataset, where the accuracy increases with increase in the number of classes. We also observed that the attacks improve in accuracy as sparsity of the lottery ticket increases and then drops. Our results demonstrate that these attacks are transferable across models with a high accuracy. We see the following research directions for the future:

- We would like to explore other pruning methods like Grasp, SNIP, in the context of membership inference attacks.

- We feel more empirical study is required into examining the relationship between attack susceptibility and generalizing capabilities of the model.

- We wish to study if the differences in the accuracy affect classes (based on their density in the training data) differently.

## Acknowledgements

We would like to thank Sharath Girish from University of Maryland for helping us with the lottery ticket code. We would also like to thank Prof. Jim Purtilo for his support during this work.

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
