# OpenReview forum: "Membership Inference Attacks on Lottery Ticket Networks"
_ICML.cc/2021/Workshop/AML — ICML 2021 Workshop AML Poster_

### Official Review · Reviewer_Sj3v · 2021-06-19
**This work is (claimed to be) the first to empirically show that the lottery ticket networks are equally vulnerable to membership inference attacks.**

**Rating:** Accept
**Confidence:** 4

**Review:**

A Membership Inference Attack (MIA) is the process of determining whether a data sample belongs to a training set of a trained model or not. Lottery Ticket Hypothesis is used to prune the networks to find smaller sub-networks that at least match the performance of the original model in terms of test accuracy in a similar number of iterations. This work shows empirical results for Membership Inference Attacks on Lottery-ticketed Networks.

Advantages: This work is (claimed to be) the first to empirically show that the lottery ticket networks are equally vulnerable to membership inference attacks.

Suggestions: It is an interesting topic. It would be of great significance to explore more about why the pruned networks are also vulnerable to adversarial attacks. More theoretical analysis may be given.

---

### Decision · Program_Chairs · 2021-06-21

**Decision:**

Accept (Poster)

**Comment:**

This paper showed that lottery ticket networks are equally vulnerable to membership inference attacks. The paper provides several interesting findings.